# Targeting the Impossible: A Review of New Strategies against Endospores

**DOI:** 10.3390/antibiotics12020248

**Published:** 2023-01-26

**Authors:** Alba Romero-Rodríguez, Beatriz Ruiz-Villafán, Claudia Fabiola Martínez-de la Peña, Sergio Sánchez

**Affiliations:** 1Departamento de Biología Molecular y Biotecnología, Instituto de Investigaciones Biomédicas, Universidad Nacional Autónoma de México, Ciudad de México 04510, Mexico; 2Centro de Investigaciones en Ciencias Microbiológicas, Instituto de Ciencias, Benemérita Universidad Autónoma de Puebla, Puebla 72592, Mexico

**Keywords:** endospore, germination, bioactive peptides, bacteriocins

## Abstract

Endospore-forming bacteria are ubiquitous, and their endospores can be present in food, in domestic animals, and on contaminated surfaces. Many spore-forming bacteria have been used in biotechnological applications, while others are human pathogens responsible for a wide range of critical clinical infections. Due to their resistant properties, it is challenging to eliminate spores and avoid the reactivation of latent spores that may lead to active infections. Furthermore, endospores play an essential role in the survival, transmission, and pathogenesis of some harmful strains that put human and animal health at risk. Thus, different methods have been applied for their eradication. Nevertheless, natural products are still a significant source for discovering and developing new antibiotics. Moreover, targeting the spore for clinical pathogens such as Clostridioides difficile is essential to disease prevention and therapeutics. These strategies could directly aim at the structural components of the spore or their germination process. This work summarizes the current advances in upcoming strategies and the development of natural products against endospores. This review also intends to highlight future perspectives in research and applications.

## 1. Endospore

Bacteria have evolved a vast arsenal for endurance and spread even under adverse conditions. Self-preservation mechanisms include scavenging, motility, and chemotaxis systems; cannibalism activation; biofilm formation; and endosporulation [1]. Indeed, there are significant advantages brought by each mechanism, and the combination of many of them, particularly in pathogens, is a critical concern for human health.

Endosporulation (indistinctly used as endosporulation or sporulation) is a complex process of morphological differentiation that culminates in the output of a dormant cell type that is exceptionally resistant to severe environmental conditions, surviving for prolonged periods without water or nutrients [2,3]. This process is highly ancient, perhaps arising near the phylogenetic root of bacteria [4,5].

The endospore lacks metabolic activity, allowing cells to survive in various environments for long periods. Wind, water, living animal hosts, and other mechanisms can spread endospores. This ability is advantageous when colonizing, prospering, and enduring a wide range of environmental conditions. Consequently, spore-forming bacteria (SFBs) occur within different ecological niches, including the soil and gastrointestinal tract of invertebrate and vertebrate animals [4,6,7,8]. However, sporulation is restricted to Firmicutes [4,6,7,8], which include the Bacilli (aerobe) and Clostridia (strict anaerobe) classes [9,10,11].

The complete set of environmental stimuli that induce sporulation is unclear, but some characterized signals include starvation [12], a viral attack [13], or abrupt oxygen change [14]. Once compromised, sporulation is irreversible and highly costly, requiring tight control to guarantee its success. In the most well-characterized sporulation model organisms, *B. subtilis* and *C. difficile*, it has been demonstrated that sporulation requires spatial and temporal coordination for the expression of hundreds of genes during spore development. The onset of sporulation is governed by the activity of conserved transcription factor Spo0A [15,16,17,18], but additional sigma factors (σ^E^, σ^F^, σ^G^, and σ^K^) orchestrate this process. Since the complete reconfiguration of the cellular architecture and metabolism is required for the formation of a mature spore, many different genes are activated during this process, including signal transduction systems, sigma factors, transcriptional regulators, metabolic enzymes, and structural proteins. For instance, at least in *Bacillus subtilis,* sporulation affects the expression of more than 500 genes [19].

A mature spore typically consists of several layers surrounding a core where genetic information is protected. (Figure 1). Among a subset of SFB species, both pathogenic and non-pathogenic, the outermost layer of the spore is the exosporium [20]. However, the typical layers in a spore are the coat, the outer membrane, the cortex, the peptidoglycan cell wall (GCW), and the inner membrane. Lastly, surrounded by all the protective layers is the core, which has a low water content and a high level of dipicolinic acid (DPA), which, together with various divalent cations and small acid-soluble spore proteins, stabilizes the DNA [7,21,22,23].

These layers of the spore have different roles in the resistance to environmental aggression and even in pathogenesis. For example, core dehydration appears to play a role in maintaining spore dormancy and resistance to high temperatures [24,25,26], while the inner membrane is a significant barrier against small molecules, possibly even water, and several potentially damaging chemicals (Figure 1) [27].

Unlike the vegetative cell wall and membrane, the inner membrane of spores has lower lateral mobility and higher viscosity [28,29,30]. In *B. subtilis*, the spore contains three germination receptors in the inner membrane, which are encoded by the tricistronic operons *gerA, gerB, and gerK*. These receptors form heterocomplexes composed of three subunits [31,32]. Other proteins located in the inner membrane are the operon-spoVA-encoded proteins, which play a role in the uptake and release of DPA during sporulation (Figure 2).

The cortex, a thick layer of specialized peptidoglycan (PG), is essential to the dehydration of the spore core and heat resistance [33,34]. Although the PG in the spore cortex is similar to that in growing cells, it has two modifications: muramic acid-lactam (MAL) and muramic acid linked only to alanine [35]. These are not present in growing cells. In addition, this layer becomes the bacterial cell wall after endospore germination.

The outer membrane is essential in spore formation but may not be a permeability barrier in mature spores. Instead, this layer is necessary for spore germination. For instance, in *B. subtilis*, GerP family channels are found in the inner membrane, allowing germinants to have access to their receptors [36,37].

More than 70 proteins compose the coat, a multilayer structure with a unique folding geometry [7,38,39] that acts as a filter, preventing the passage of large molecules (Figure 1). A consequence of this constraint is that structures within the spore are protected from degradation enzymes such as lysozyme, which readily lyses the cortex [21,38,40]. Given its biochemical complexity, it is likely that the coat has other functions, although these are still poorly understood [21]. Interestingly, only 25% of the more than 70 proteins that compose the spore coat are conserved between *B. subtilis* and *C. difficile* [7], which suggests that the spore surface may be a significant source of evolutionary adaptation [41].

In some bacteria, such as *B. cereus* and *C. difficile*, a layer of protein, the exosporium, surrounds the coat. The exosporium layer is a highly diverse and complex structure, which is not present in all endospore-forming strains. It is a barrier against penetration by large molecules but still permeable to small molecules that are required for germination. Furthermore, the exosporium also confers resistance to chemicals such as ethanol, toluene, chloroform, phenol, and nitric oxide [21,42]. Generally, the exosporium is composed of a paracrystalline basal layer and an external layer consisting of a nap of fine filaments, termed the hairy nap. In *C. difficile*, this external hair-like nap is principally composed of collagen-like glycoprotein BclA [43], which appears to be involved in adherence and entry into the host [44].

Spores assemble within the mother, where genome segregation and asymmetric cell division occur [45,46,47]. It begins with the formation of a polar septum, resulting in two unequal cells, a mother cell and the forespore. They have identical copies of the chromosome, but express distinct genetic programs controlled by specific sigma factors [45,46,47]. The mutation in all sigma factors abrogates spore formation [41,45].

After polar division, the forespore is engulfed by the mother cell. In late stages of development, it directs the assembly of the cortex, where several multiprotein layers encase it [40,48,49,50,51]. When the spore reaches maturity, the mother cell lyses, freeing it into the environment where it persists until the conditions are suitable for germination (Figure 2) [28,30].

Germination is a complex mechanism by which the metabolism and macromolecular synthesis are restarted to give rise to a new vegetative cell. This process involves, during dormancy, the ability of spores to sense their environmental situation, and once the conditions are suitable for resuming growth (generally, specific nutrients in the environment), germination occurs. During germination, the resistance properties are lost [52,53,54] (Figure 2). The dormant spore is transformed into a metabolically active vegetative cell upon the binding of the germinant and, in some cases, co-germinants to their appropriate receptors. Upon the binding of the germinant to its receptors on the spore inner membrane, the signal is transduced, resulting in the activation of proteins that allow small molecules to move across the membrane and deconstruct the protective layers, restoring regular hydration and active metabolism [54].

Germinants are low-molecular-weight biomolecules found in the environment where the growth of the organism is favored. Common germinants may include amino acids, sugars, purine nucleosides, inorganic salts, or combinations of these molecules [28,55]. It is also possible to induce spore germination through a receptor-independent process, including exogenous CaDPA, lysozyme, dodecylamine, or extremely high pressure or heat. In addition, fragments of vegetative cell peptidoglycan induce spore germination in a protein kinase-dependent manner that is not at all understood [30,54]

Germination is defined as events beginning at the time of germinant recognition and ending when the water content in the spore core increases to 80% of wet weight [30]. This water content allows normal metabolism and macromolecular synthesis for outgrowth to occur. Five main events have been described in spore germination: germinant sensing, commitment to germinate, release of CaDPA, hydrolysis of cortex peptidoglycan by cortex-lytic enzymes, and finally, swelling and growth (Figure 2) [30]. The last stage of germination involves core swelling, germinal cell wall remodeling, internal membrane remodeling, internal membrane lipid mobility, and core protein mobility [30]. At the end of germination, cells take their usual rod shape and get ready to divide (Figure 2).

## 2. Spore Resistance and Killing Procedures

As previously described, the layers of the spore and core characteristics are responsible for their extensive resistance to varying ranges of temperature and pressure, ultraviolet radiation, and many harmful chemical substances such as hypochlorite and aldehydes. The resistance properties of spores to various agents may differ among strains, species, and genera, which can impact protocols for food sterilization, instrumental and surface sanitization, and even pharmacological treatments.

Despite its resistance properties, some extreme treatments can damage different components of the spore, including DNA, inner membrane, and proteins within the spore core. This can lead to the death of the spore. There are numerous excellent reviews [29,56,57,58,59] on different aspects for spore eradication. Here, we summarize some of the most used techniques (Table 1). In the upcoming sections, we will further analyze novel technologies and their applications. It is important to note that the resistance mechanisms of eradication techniques are mostly studied in *B. subtilis*. Further information is needed on other genera and species.

## 3. Pathogen Spore-Forming Bacteria

As mentioned, all SFBs are members of the Firmicutes phylum, but only some members of Firmicutes make spores. SFBs can be found in four different classes of Firmicutes, Bacilli, Clostridia, Erysipelotrichia, and Negativicutes, all of which encode similar sets of core sporulation proteins and low G+C content. It has recently been proposed that sporobiota and sporobiome indicate the members and genomes of these types of bacteria [87]. Since spores are resistant to traditional DNA isolation methods, they may be underrepresented in metagenomic studies. As a result, we have a limited understanding of their types and abundances [88].

Among these SFBs, only a few have been associated with disease. Examples are the Clostridiaceae family, causing severe affections such as tetanus, gas gangrene, and botulism (Table 2) [89], and some Bacilli classes, such as *Bacillus anthracis* and *Bacillus cereus sensu lato*. The most common feature of the pathogenic *Clostridia* and *Bacilli* is the cell and tissue damage that they cause primarily due to the production of potent extracellular toxins (Table 2). Furthermore, some of these microorganisms have motility by flagella (swimming and swarming, e.g., *B. cereus*) or by type IV fimbriae (gliding motility, e.g., *Clostridium perfringens*), which leads to biofilm formation and antibiotic resistance promotion. The ability of these bacteria (e.g., *C. perfringens*) to form biofilm facilitates gastrointestinal infections and their persistence despite antibiotic treatments [90]. Additionally, it is necessary to avoid human and animal diseases and food contamination by spore formers to avoid prolonged antibiotic treatments. In addition, limiting antibiotics in farms to avoid multiresistant pathogen spore formers (PSFs) in this group of particularly challenging bacteria, considering the inherent resistant properties of the spore and its high rates of transmission and dissemination.

Proper handwashing is a critical step to prevent infections and spore dissemination. In addition, it is a crucial practice for controlling nosocomial infections, including *C. difficile* and *B. cereus* infections [91,92,93]. Therefore, hospital staff and the community should be reminded of this and trained in hand hygiene techniques.

**Table 2 antibiotics-12-00248-t002:** Main human spore-forming pathogens.

Disease	SFP	Target	Toxins	Remarks	References
Anthrax	*B. anthracis*	Cutaneous, gastrointestinal, and pulmonary infection	Tripartite anthrax-toxin	Anthrax is endemic in several regions around the world, and its epidemiology mainly depends on its dynamics in wildlife, local agriculture, community education on transmission routes, and access to health and vaccination.	[94,95,96,97]
Food poisoning	*B. cereus*	Gastrointestinal system: diarrheal and emetic syndrome	Hbl, Nhe, CytK, and cyclic peptide c cereulide	Both syndromes, gastrointestinal and emetic, are generally mild and self-limiting. However, severe, and even lethal cases of emetic foodborne *B. cereus* disease have been reported.	[8,98]
Botulism	*C. botulinum,* *C. baratii, and* *C. butyricum*	Gastrointestinal, nervous, and muscular systems	Botulinum neurotoxin (BoNT)	Metabolic and biochemical tests have divided the *C. botulinum* strains into four groups, while antibody neutralization has separated the neurotoxins into seven serotypes.	[99,100,101,102,103,104]
Food poisoning, myonecrosis (Gas gangrene), fatal infections in postdelivery women, and necrotizing colitis	*C. perfringens*	Gastrointestinal system, and wound and extremities	α-, β-, ε-, and ι toxins are the most common toxins	Currently, 23 virulence genes that encode toxins and virulent enzymes have been identified in *C. perfringens*, making it the most prolific toxin-producing pathogen presently known.	[105,106,107]
*C. tetani*	Tetanus	Nervous and muscular systems: local, cephalic, and neonatal tetanus	Tetanospasmin (also called tetanus neurotoxin; TeNT)	People who have not been vaccinated are more likely to have cases of generalized tetanus, while people who are poorly immunized are more likely to have local cases. With this disease, the most enduring challenge has been the prevention measure of free vaccination campaigns.	[108,109,110,111]
Pseudomem-braneous colitis	*C. difficile*	Gastrointestinal system (colon)	Toxin A (TcdA) and toxin B (TcdB)	The fecal–oral route transmits spores. These bacteria colonize the large intestine when there is dysbiosis in the gut microbiota caused by antibiotic treatments. It is associated with multiple relapses and recurrence.	[112,113,114]
Septic shock and necrotizing fasciitis	*C. sordelii*	Mostly associated with gynecological complications in women	Pathogenic strains of *C. sordellii* generate up to 7 identified exotoxins; among them, the lethal toxin (LT) and hemorrhagic toxin (HT) are regarded as the major virulence factors	The infection progresses rapidly. Thus, therapeutic interventions are rarely successful. At present, there is no antitoxin available.	[89,115,116]

## 4. Approaches in the Food Industry to Control of Spore-Forming Bacteria 

For several reasons, aerobic and anaerobic SFBs are a critical concern in the food industry. First, the widespread distribution of spores makes it impossible to prevent their presence in raw food and ingredients. As previously described, spores are extremely resistant to heat, dehydration, and chemical or physical stresses. Typical treatments used in the food industry, including heat treatments, can inactivate vegetative cells but fail to kill spores. Survivors may germinate and proliferate rapidly in the product or in the intestinal tract leading to food spoilage or active infections. Consequently, intensive wet heat treatment, generally at a temperature higher than 100 °C, is usually applied to inactivate spores in food products. Therefore, the biggest challenge in food production is to prevent its contamination, especially by SFBs, while avoiding the loss of organoleptic characteristics. Another critical point is people’s awareness of the proper storage of food. Storage at the wrong temperature can lead to contamination, for example, by *C. botulinum*, which can lead to poisoning and death. Different types of labeling have been proposed to prevent this [104]. 

Food spoilage SFBs are typically Bacillales, including the *Bacillus*, *Geobacillus*, *Anoxybacillus*, *Alicyclobacillus*, and *Paenibacillus* genera, while among *Clostridiales*, cases of contamination by species of the *Clostridium* and *Desulfutomaculum* genera have been reported [117].

SFBs become especially relevant in powdered dairy products (whole and skimmed milk, whey and milk isolates, whey and milk protein concentrate, casein, and caseinates) [118]. These bacteria can be controlled with physical treatments, such as dehydration, low-temperature storage, pasteurization, thermal sterilization, and nonthermal methods, such as irradiation and ultra-high-pressure processing [119]. High-temperature heating followed by microfiltration is a common method for milk treatment [120], but this procedure often leads to the development of psychrotolerant SFBs, mainly due to re-contamination [120].

Another food product marketed in powder form is dry sardine meal in Japan [65]. If the short-time ultra-pasteurization method is followed by rapid cooling to 60 °C, it allows the destruction of spores with a minimum change in the product color to be achieved [65].

Supercritical fluids (SCFs) and cold plasma are other techniques that do not alter the sensory characteristics of food products [78]. Plasma is generated when a gas is subjected to an electric current. It consists of a wide range of molecules and atoms in an excited state, such as ions, electrons, free radicals, reactive species (reactive oxygen and nitrogen species), and UV radiation [79]. The mechanism of action is membrane destruction by free radicals and induction of oxidative stress that produces the oxidation of enzymes and lipids [80]. The disadvantages are lipid oxidation in fish or decomposition of oligosaccharides in juice [79].

A supercritical fluid (SCF) is a substance at a temperature and pressure above its critical point where there is equilibrium between gas and liquid. The SCF shows gas-like viscosity, intermediate diffusivity, and liquid-like density, which provides it with good penetrability [83]. Carbon dioxide has been used for sterilizing food products, since it is considered GRAS (because it is environmentally friendly and non-corrosive), has good solubility, and is easy to remove and recycle. Hart et al., 2022, observed that SCF-CO2 was effective in inactivating bacterial spores without modification of the properties of the food [82]. The proposed mechanism of action is that SCF-CO2 destroys the spore cell wall, coat, cortex, and membrane. In the case of the vegetative cells, SCF-CO2 destroys the cytoplasmic membrane and degrades proteins, and altered pH is observed, since the fluid interacts with the cellular components, producing a loss of enzymatic activity [81]. Even though there are patents for the application of SCF-CO2 to food, there are still no processes at an industrial level.

Germination elimination at high isostatic pressure does not alter the sensory characteristics of food products. This strategy involves artificial induction or germination using chemical or physical media followed by a mild treatment. The spore loses its resistance properties upon germination induction and can be killed with less aggressive techniques. However, a critical disadvantage is that it allows superdormant spores to survive [84]. In various germination elimination processes, superdormancy of spore subpopulations has been observed. Superdormancy is a term that describes a group of spores that is different from the rest of the population in terms of their ability to germinate. These spores remain dormant or germinate extremely slowly [86].

Currently, the exact causes of spore superdormancy are unclear, but it is mainly explained by two factors. One factor is heat activation, which reverses spore germination; the other is related to the spore’s low levels of germinant receptors. Some research groups are studying the characteristics of superdormant spores to design novel strategies that prevent this subpopulation’s presence and reduce the potential spoilage of food products [84].

## 5. Natural Products against the Spore

Natural products are “small molecules produced by any organism, including primary and secondary metabolites”. Natural products can have simple chemistry, as in the case of urea, or complex structures, as in the case of Taxol. Generally, these natural molecules are only obtained in small quantities and may have a wide range of biological activities (https://www.nature.com/subjects/natural-products, accessed on 24 October 2022). As mentioned before, the pursuit of developing new procedures to combat spores in food products is a very active area. Furthermore, new treatments to combat human infections caused by SFPs are also an active area of research. Various natural products have been studied as promissory strategies to target endospores. Particularly, the microbial natural products bacteriocins have demonstrated activity against germinated spores, i.e., the state in which the spore inner membrane is exposed (Table 3).

The most well-studied bacteriocin is nisin, a 34-amino-acid-residue polypeptide produced by strains of *Lactococcus lactis* that is inhibitory toward many pathogens [121]. Nisin inhibits Gram-positive vegetative cells by generating membrane pores and interfering with cell wall biosynthesis by interacting with lipid II [71]. Furthermore, recent super-resolution structured illumination microscopy (SR-SIM) analysis showed the condensation of chromosomal DNA in *Staphylococcus aureus* cells exposed to nisin, suggesting that nisin interferes with chromosome replication or segregation in *S. aureus* [72].

Although spores differ from vegetative cells, nisin prevents the outgrowth of spores from *Bacillus* and *Clostridium* species (Table 3) [85,122]. Studies to understand the mechanism of action of nisin have shown that nisin mutants in the hinge region can bind lipid II but are incapable of forming pores and are active against *B. anthracis* vegetative cells without affecting spore outgrowth [122]. This evidence suggests that inhibitory mechanism of nisin acting on the outgrowth of *B. anthracis* spores is a combination of binding to lipid II and membrane disruption (pore formation) [122]. Furthermore, recent observations have shown that nisin treatment of *B. subtilis* spores caused the inner membrane of germinated spores to appear smaller than that of untreated spores [123].

Due to its broad spectrum against Gram-positive bacteria, including foodborne pathogens, the nisin effect has also been tested on nosocomial pathogen *C. difficile.* The vegetative form is also susceptible to the nisin effect in vitro [124] and also in an ex vivo model of the colon [125]. One benefit of nisin, and probably many other bacteriocins, is the ability to destroy *C. difficile* with minimal impact on the microbiota composition [125]. High nisin concentrations (25.6 μg ml^− 1^) reduced spore viability by 40–50 %, inhibiting *C. difficile* vegetative growth and its germinated spores [124]. Nisin also affects germinated spores of *C. perfringens* and other clostridia (Table 2). Interestingly, germinated spores of *C. perfringens,* depending on their origin (food poisoning (FP) or non-foodborne (NFB)), exhibit differential nisin resistance, with the highest being in NFB isolates [126]. These results may reflect changes in the spore layer composition, as has been observed in the spore resistance of *C. botulinum* species [127].

Other bacteriocins can also inhibit spore outgrowth and germination or diminish the heat resistance of different spores (Table 3). To be active, most of these peptides require the germination of spores or mixed treatments at high temperatures or pressure, but some molecules seem to be active on the resting spores of some species. For instance, Plantaricin causes critical damage to the morphology of cells and spores of *B. cereus* [128,129]. This bacteriocin removes the exosporium, and the spores look hollow [128,129]. Enterocin ASK48 affects the endospore structure without any pre-germination treatment. The exposure of *Alicyclobacillus acidoterrestris* to Enterocin ASK48 results in the disorganization of its endospore structure [130].

As inferred, a single bacteriocin can have heterogeneous effects depending on the origin of the spores (Table 3). Thus, some spores appear more “fragile” or susceptible to specific bacteriocins. Furthermore, these heterogeneous outcomes seem to depend on the bacterial species involved, since the same bacteriocin can affect spore ultrastructural characteristics differently.

One critical observation about these natural products is that even when the majority do not kill spores, these peptides have the sporostatic effect of preventing spore outgrowth; therefore, the proliferation of the vegetative forms and production of lantibiotics and other peptides do not confer resistance as conventional antibiotics do. However, nisin-resistant mutants of *C. botulinum* appeared through continuous exposure to the lantibiotic, causing their ability to germinate at levels of nisin that typically reduced their parental strain [131,132]. Although the mechanism that can bypass nisin action in these resistant mutants is unknown, it is not a nisin-specific phenomenon, as resistance can also be observed in various bacteriocins across a range of classes [131]. The bacteriostatic effects of nisin and other bacteriocins are a huge disadvantage for massive utilization in products where sterility is critical.

**Table 3 antibiotics-12-00248-t003:** Bacteriocins against endospores.

Bacteriocin	Producer	Spore Tested	Remarks
Haloduracin	*Bacillus halodurans*	*B. anthracis*	It inhibited spore outgrowth [133]
Nisin	*Lactococcus lactis*	*C. perfringens*	It arrested outgrowth of germinated spores in rich medium, but it did not affect a meat model system [126]
		*C. sporogenes*	Effective when spores germinated but not on spores themselves [134]
		*C. difficile*	At high concentrations, nisin appeared to cause a statistically significant decrease in the viability of non-germinated spores [124]
		*C. beijerinckii*	Active with previous high temperature [124]
		*C. botulinum*	Heated spores were very sensitive to nisin, and nisin-treated spores became more heat sensitive; nisin acted as a pro-germinant [135]
		*B. subtilis*	Sporicidal on germinated spores [123]
		*A. acidoterrestris*	At high concentrations ranging from 0.1 to 1.5 mg liter^−1^, nisin had an inhibitory effect without the application of any previous thermal treatment [136].
Enterocin ASK48	*Enterococcus faecalis*	*A. acidoterrestris*	Active on resting spores [137]
Enterocin EJ9	*Enterococcus faecalis EJ97*	*Geobacillus stearothermophilus*	Heat-activated endospores became sensitive [138]
Bificin C6165	*Bifidobacterium animalis* subsp*. animalis CICC 6165*	*A. acidoterrestris*	In commercial diluted apple juice, no significant activity of bificin was observed against the endospores, but its addition contributed to the reduction in thermal resistance [139]
Lacticin	*L. lactis* IFPL 3593 *two-peptide lantibiotic*	*C. tyrobutyricum*	It inhibited the germination of clostridia spores and decreased the number of clostridia spores [140]
Plantaricin	*Lactobacillus plantarum TF711*	*C. sporogenes*	Clostridia spore count was significantly lower in the experimental cheese model [141]
Plantaricin JY22	*Lactobacillus plantarum JY22*	*B. cereus*	It affected spore integrity; the exosporium was peeled out, and leaking spores were hollow [129]
Plantaricin YKX	*L. plantarum*	*Alicyclobacillus* spp.	Bacteriocins could induce the germination of *A. acidoterrestris* spores [142]; in endospore suspensions, cell viability decreased in proportion to the bacteriocin concentration added [130]
Thurincin H	*Bacillus thuringiensis*	*B. cereus*	Decrease in viable counts only when germination was induced by BHI (*p* < 0.05) [143]

## 6. Conclusions and Future Directions

Spores are associated with persistence and resistance, and they are difficult to eliminate from food, surfaces, or even human bodies. The eradication of spores from surfaces and food is critical, since spores not only can affect the quality of food products, but they can also serve as vectors for different diseases, including potentially fatal ones. It is also difficult to eradicate an infection when one occurs in a person. Infections are associated with multiple relapses, reinfections, and further pathogen dissemination. Therefore, it is critical to prevent infection by assuring food security and by breaking the contagious chain with the proper sanitation of surfaces and handwashing. Particularly relevant for health workers dealing with opportunistic infections, such as *C. difficile*, is to prevent pathogen dissemination. Although vaccines have been developed to counteract infections by spore formers, some countries are still behind in their vaccination programs. This makes people living in those countries more vulnerable to infections. As seen in the COVID-19 pandemic, health is a concern of all countries. As the One Health program declares, all is interconnected. Therefore, it is crucial to ensure food safety and global health, especially considering that spore formers are less restricted by physical and temporal barriers and are thus easier to disperse.

Heat and chemicals have been used extensively for spore eradication, particularly for material and surface decontamination. However, when dealing with delicate substances and materials, such as food products, there are fewer options available for spore eradication. Furthermore, treatments for infections by spore-forming pathogens are very limited, and now, the global problem of antimicrobial resistance makes these infections even more challenging to treat. Traditional methods for spore elimination in food products have the great disadvantage of affecting the organoleptic properties or have raised concerns about biosecurity, as the g-radiation method has. Plasma and supercritical fluids are promising methods for eliminating spores from food without affecting the organoleptic properties, but these methods are expensive and limited to small-scale procedures. The use of small peptides such as bacteriocins is not only possible in food but also as pharmacological treatment for infections. Unfortunately, most of the cases require pre-activation (induction of germination) treatment. Besides the high cost associated with germination procedures, heterogeneous spore germination and superdormancy are concerns when using germination eradication procedures. Consequently, the disruption of the formed spores existing in delicate materials or causing infection in human beings remains a considerable challenge. With a better understand of endospore physiology, especially the properties of endospore layers, we could design more specific strategies for their control and eradication.

## Figures and Tables

**Figure 1 antibiotics-12-00248-f001:**
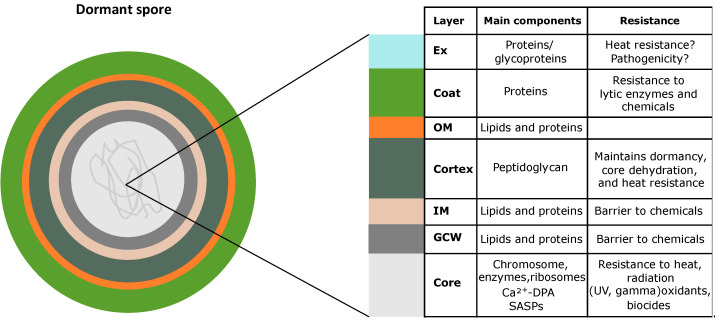
The spore is composed of a series of concentric layers contributing to its resistance properties. Abbreviations: Ex, exosporium; Ct, coat; OM outer membrane; Cx, cortex; IM, inner membrane; GCM, germ cell wall.

**Figure 2 antibiotics-12-00248-f002:**
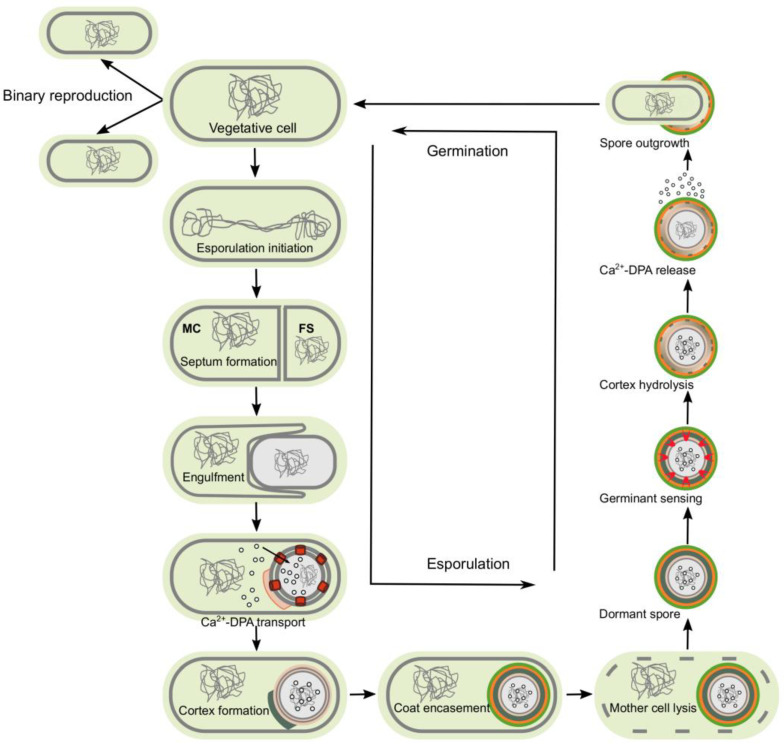
Sporulation begins once the cell senses unfavorable environmental conditions (e.g., starvation). The first structural event is the formation of a polar septum, which divides the cell into two asymmetric compartments, the mother cell (MC) and the forespore (FS). Next, after the completion of DNA segregation, the mother cell engulfs the forespore. Calcium dipicolinic acid (Ca-DPA) is synthesized in the mother cell and transported into the forespore in exchange for water. The forespore is then coated with the cortex and coat layers. Finally, once the spore is mature, the mother cell lyses, releasing the dormant endospore into the environment. Germination is activated upon sensing an appropriate small molecule, such as amino acids. The dormant spore initiates a signaling cascade that activates hydrolases and core hydration, which is necessary for metabolism to resume in the germinating spore.

**Table 1 antibiotics-12-00248-t001:** Spore eradication techniques.

Agent	Remarks	Target	Resistance Mechanism	Application	Disadvantages	References
UV radiation	UV 254 nm radiation is the most effective in killing spores; potentially, UV 222 nm	DNA	SASPs, DNA repair enzymes, and coat	Surface decontamination	Mutagenic effects of UV 254; few dosage/efficacy data	[57,58,60,61,62,63]
γ-Radiation	Cobalt-60 (Co60) and/or cesium-137 are generally used for gamma-radiation procedures, and 3 kGy was the lowest and 30 kGy the highest doses adequate for the inactivation of 10^6^–10^8^ spores	DNA	SASPs and DNA repair enzymes	Food processing applications, medical devices and pharmaceutical products, and maybe powdered foods	Specialized and expensive equipment: concern about biosecurity of γ-irradiated foods	[62,64,65,66]
Dry heat	Can be conducted with hot air using a forced convection oven; standard treatments are 160 °C for 2 h or 170 °C for 1 h	DNA	SASPs	Glass and metal medical devices, and materials of spacecraft hardware	Incompatible for several materials (plastic and rubber): spores of some bacteria are resistant to dry heat; cost of electricity	[58,67,68,69,70,71]
Wet heat	Refers to environments at elevated temperatures and saturated with moisture (100% RH) or boiling water	Inactivation of core enzymes	Low water content and SASPs	Medical/surgical supplies, microbiological growth media, and hospital waste	Material incompatibility Incorrect heat treatment could also generate heterogeneous spore germination; spores of some species are resistant to ≥100 °C	[34,72,73]
Chemicals	Oxidizing agents, aldehydes, and acids and alkali disinfectants	It depends on the chemical nature of the agent but may include: DNA, inner membrane, germination enzymes, and core proteins	SASPs, coat, low permeability	Hospital and industry facilities, and surfaces	Toxic and corrosive effects; residue generation	[57,74,75,76,77]
Plasma		DNA damage	Further studies are needed	Packing material and food	Food oxidation; plasma source is relatively small and must be in proximity	[58,78,79,80]
Supercritical fluids	Supercritical CO2	Spore cell wall, coat, cortex, and membrane	Further studies are needed	Packing material and food	Small scale	[81,82,83]
Germinants	Induction of germination with enzymes, nutrients, or heat and pressure				Expensive; heterogeneous germination; superdormancy	[67,84,85,86]

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
