# Peer review of "Targeting the Impossible: A Review of New Strategies against Endospores"

_antibiotics, 2023, doi:10.3390/antibiotics12020248_

Round 1
Reviewer 1 Report
The article “Targeting the impossible, new strategies against endospore” is exciting and potentially very interesting for the microbiology community. This manuscript explores current knowledge on spore formation and the elimination of spores from food products. Unfortunately, the lack of organization in the paragraphs and the limited command of English grammar limits its value. Furthermore, the article does not have a final section closing section, and it ends abruptly. It requires “future directions” of research and what the industry and food science will bring to curtail food spoilage and disease. I suggest a re-write and resubmission.
Author Response
The authors appreciate the critical comments raised by the reviewer. As a result, we have rewritten the text in several sections and made extensive changes to improve clarity and language style. We cannot cite specific lines because there are several changes across all sections. We are attaching a marked document where all changes are tracked.

Reviewer 2 Report
Please refer to the attached document.

Author Response
The authors appreciate the reviewer critical comments and suggestions on this manuscript
1) Title of the article needs more clarity. Please specify that the manuscript is a review article. The title was modified to clarify the issue as follow: Targeting the impossible, a review of new strategies against endospore
2) Although the authors have described various agents used against spores, the heat resistance of spores needs more elaboration. The effect of physical agents like heat, moist heat and their role in the eradication of spores from surgical equipment’s need to be mentioned.
Response. We agreed, and a new section (Spore resistance and killing procedures) was introduced in the text. This section summarizes standard techniques used against spores.
3) Hand hygiene is still the most effective method for control of Clostridoides difficile. The usefulness of handwashing with soap and water need to be included among the existing methods for control of spore forming medically significant bacteria.
Response. We introduced the critical importance of handwashing in the “Pathogen Spore-forming bacteria” section. Please refer to lines 179-181.
4) It needs future studies to prove the in vivo and in vitro efficacy of bacteriocins like nisin. Their bacteriostatic properties might limit their use in areas demanding completely sterile environments. Such limitations should be mentioned in the discussion part.
Response. We mentioned limitations of bacteriocins in lines 288-290.
Furthermore, we are attaching a new document where all changes are tracked.

Round 2
Reviewer 2 Report
Necessary changes have been made and is satisfactory.